# In Vitro Studies on the Antimicrobial and Antioxidant Activities of Total Polyphenol Content of *Cynara humilis* from Moulay Yacoub Area (Morocco)

**DOI:** 10.3390/plants11091200

**Published:** 2022-04-29

**Authors:** Mostafa El Khomsi, Mohammed Kara, Anouar Hmamou, Amine Assouguem, Omkulthom Al Kamaly, Asmaa Saleh, Sezai Ercisli, Hafize Fidan, Driss Hmouni

**Affiliations:** 1Natural Resources and Sustainable Development Laboratory, Department of Biology, Faculty of Sciences, Ibn Tofail University, B.P.133, Kenitra 14000, Morocco; Hmouni.driss@uit.ac.ma; 2Laboratory of Biotechnology, Conservation and Valorisation of Natural Resources (LBCVNR), Faculty of Sciences Dhar El Mehraz, Sidi Mohamed Ben Abdellah University, BP.1796 Atlas, Fez 30000, Morocco; 3Laboratory of Engineering, Molecular Organometallic Materials and Environment, Faculty of Sciences Dhar El Mehraz, Sidi Mohamed Ben Abdellah University, BP.1796 Atlas, Fez 30000, Morocco; anouar.hmamou@usmba.ac.ma; 4Laboratory of Functional Ecology and Environment, Faculty of Sciences and Technology, Sidi Mohamed Ben Abdellah University, B.O. Box 2202 Imouzzer Street, Fez 30000, Morocco; assougam@gmail.com; 5Department of Pharmaceutical Sciences, College of Pharmacy, Princess Nourah bint Abdulrahman University, P.O. Box 84428, Riyadh 11671, Saudi Arabia; omalkmali@pnu.edu.sa (O.A.K.); ASAli@pnu.edu.sa (A.S.); 6Department of Horticulture, Atatürk University, 25240 Erzurum, Turkey; sercisli@gmail.com; 7Department of Tourism and Culinary Management, Faculty of Economics, University of Food Technologies, 4000 Plovdiv, Bulgaria; hfidan@abv.bg

**Keywords:** *Cynara humilis* L., antimicrobial activity, antioxidant activity, in vitro, total polyphenols, 2.2-diphenyl-1-picrylhdrazyl

## Abstract

In Morocco, *Cynara humilis* L. is used in traditional medicine. The objective of this research was to research the antioxidant and antimicrobial properties of hydroethanolic extracts from the *C. humilis* plant’s leaves and roots. The content of polyphenols and flavonoids was evaluated using Folin–Ciocalteu’s and aluminum chloride assays. Two techniques were used to evaluate antioxidant properties: antioxidant capacity in total (TAC) and 2,2-diphenyl-1-picrylhdrazyl (DPPH). In antimicrobial assays, five pathogenic microbial strains were studied including two *Escherichia coli*, one coagulase-negative *Staphylococcus* and *Klebsiella pneumoniae*, and one *Candida albicans*, by two techniques: agar disk diffusion and microdilution. Leaves had a greater content of flavonoids 27.07 mg QE/g of extract and the polyphenols 38.84 mg GAE/g of extract than root 24.39 mg QE/g of extract and 29.39 mg GAE/g of extract, respectively. The TAC test value of the 0.77 mg AAE/g extract in the leaf extract was found to be significantly greater than that of the 0.60 mg EAA/g extract in the root extract. The DPPH antioxidant assay IC_50_ values of the root and leaf extract were 0.23 and 0.93 µg/mL, respectively. *C. humilis* extracts showed an antimicrobial effect against all tested strains, the inhibitory zone (DIZ) have values in the range between 12 and 15 mm. Moreover, the root extract showed the lowest minimum inhibitory concentration (MIC) against coagulase-negative *Staphylococcus* with an IC_50_ value of 6.25 mg/mL. The higher content of flavonoids and polyphenols in the hydroethanolic extracts of *C. humilis* leaves and roots demonstrates that they have a significant antimicrobial and antioxidant effect, as found in this study.

## 1. Introduction 

The therapeutic value of medicinal and aromatic plants has long been recognized, and the concept that plants contain pharmacologically active substances has been well defined. Plants have been used to cure disorders of microbial origin as well other disorders since ancient times. The first healers were priests or doctors. Indeed, science and religion have been associated with illness and healing for more than 20,000 years, and their goal has always been to heal the soul and the body [1,2,3,4,5]. A large number of active ingredients used in the preparation of medicines are of plant origin [6,7]. Indeed, scientific research has documented that the extracts of medicinal and aromatic plants and essential oils coming from many plants have medicinal properties that can be used in the phytotherapy, especially antibacterial, antioxidant, antifungal and also pharmacological properties [8,9,10].

Antioxidants used to fight free radicals can also be used to cure of certain illnesses, including heart problems, diabetes, neurodegenerative diseases and also cancer [11,12]. The use of synthetic antioxidants can be the cause of many diseases, such as carcinogenicity, which makes peoples tend to use antioxidants of natural origin, especially of plant origin, which represents a very important source of natural antioxidants [13]. The antimicrobial activity of natural antioxidants is well reported in the literature, and many natural chemicals of plant origin have been shown to have antimicrobial activities [14,15,16]. The biological compounds responsible for these properties are the secondary metabolites present in plants, such as polyphenols, flavonoids, terpenes, coumarins, tannins, and alkaloids [17,18,19]. For this reason, scientific research must be applied to the search for new plants to study their phytochemical composition and also their biological and pharmacological activities [20].

*Cynara humilis* L. is a tiny plant from the Mediterranean region and belongs to the Asteraceae family [21]. Based on genetic and morphological data, *C. humilis* was part of the secondary gene pool of *C. cardunculus* [21,22,23]. In Morocco, the roots of *Cynara humilis* L are traditionally used to treat wounds and burns [1,24,25]. The healing effect of the plant is well demonstrated in previous research [26]. In addition, because of its choleretic activities, the plant is used in therapy to increase bile secretion [27]. Other studies have indicated that *C. humilis* has been used in cheese making [28,29]. Phytochemical studies of methanolic extracts of the *C. humilis* plant have documented the presence of phenolic compounds, flavonoids, coumarins, and aurone [27]. Other research has documented that the biological activities of the plant have not yet studied [26].

However, no study has documented the plant’s antimicrobial and antioxidant activities to our knowledge, and scientific research on this plant is few and limited. Within this framework, our research is the first to delve into the antioxidant and antibacterial biological characteristics of *C. humilis* root and leaf extracts, and also to evaluate its total polyphenol and flavonoid composition. 

## 2. Results and Discussion

### 2.1. Yield of Extraction

The yield of extraction of the roots and leaves of *C. humilis* showed that the extract of the leaves presented a yield 17.4% higher than that of the extract of the roots, which presented a yield of 12.8% (Table 1). 

### 2.2. Phenolic Compounds and Flavonoid Contents

Polyphenols and flavonoids are types of secondary metabolites found in almost all plants, and their quantity varies depending on geographical origin, species, and variety [30,31,32]. The characteristic structure of each of these aforementioned compounds indicates a specific behavior/action/role, as statistically demonstrated in [33]. It is difficult to judge the global composition of polyphenols in a single study carried out in spring on a single site. The results of the polyphenol assay are written in unit (mg GAE/g of extract) which means milligrams of gallic acid equivalent by gram of extract) (Figure 1). The two extracts tested, leaves and roots, showed a statistically significant difference (*p* = 0.04). Leaves represented a value of 38.84 ± 5.15 mg GAE/g extract, compared to roots with 29.56 ± 1.44 mg GAE/g extract. When compared to the findings of previous investigations, our results show that the quantity of polyphenols in *C. humilis* is higher than that of *C. cardunculus*, which varied between 6.96 and 14.79 mg GAE/g extract [34], and lower than that of *C. scolymus*, at 50 mg GAE/g extract [35]. Several environmental factors, including climatic conditions, geographical location, soil fertility, cultivar genotype, and experimental factors, including the part of the plant used, the time of harvesting, the extraction method, as well as the polarity of the solution being used, and extraction time, can influence phenolic compounds in plants [36,37,38,39]. HPLC chromatographic analyses have shown that the *C. humilis* plant contains numerous phenolic compounds including isochlorogenic acid, chlorogenic acid and caffeic acid [27].

The quantity of flavonoids in plant extracts was evaluated using the quercetin calibration curve, y = 0.0062 + 0.4289 and *R*^2^ = 0.857, flavonoid results presented by the unit (mg QE/g of extract) which means milligrams of quercetin equivalents by gram of extract. The flavonoid values in the leaf extract (27.07 ± 2.79 mg QE/g of extract) are superior to those found in the root extract (24.39 ± 2.87 mg QE/g extract). The analysis of these finding revealed no statistically significant differences (*p* = 0.309) (Figure 1). In the literature, the biological activities, especially antioxidant and antibacterial properties, have been widely documented [14,40]. A phytochemical study indicated the presence of the following flavonoids, luteolin, apigenin, quercetin, maritimen in the *C. humilis* plant [27].

### 2.3. Antioxidant Assay

#### 2.3.1. Antioxidant Capacity Total (TAC)

An ascorbic acid calibration curve was created to determine the overall antioxidant capacity, and the value were represented by the unit (mg EAA/g of extract) which means milligrams of ascorbic acid equivalent by gram of extract. the leaf extract has 0.77 ± 0.07 mg AAE/g extract total antioxidant capacity, is more effective than the root extract, which had a TAC of 0.60 ± 0.04 mg AAE/g of extract, as per TAC test findings (Table 2). The difference of the two extracts analyzed was found to be significant (*p* = 0.023) in the results analysis.

#### 2.3.2. DPPH Free Radical Scavenging Antioxidant Activity

Polyphenols and flavonoids are abundant in the *C. humilis* plant, and these compounds are renowned for their antioxidant properties, scavenging of free radical by inhibiting the traces of metal ions that participate in their formation by exchange of proton atoms [30]. The IC_50_ value is the necessary concentration to lower DPPH by 50%. According to Table 3, the IC_50_ values of the investigated substances differed by a significant amount (*p <* 0.05). The results obtained indicate that leaves and roots of the plant had an inhibitory concentration of 0.23 ± 0.02 and 0.93 ± 0.01 μg/mL, respectively, which is higher than those of the reference antioxidants; quercetin, BHT, and ascorbic acid had a concentration of 0.05, 0.20, and 0.16 ± 0.01 μg/mL, respectively. As a lower IC_50_ value indicates a higher antioxidant capacity, this shows that both extracts had lower antioxidant activity than the reference antioxidants. The phenolic content among those sample explains the antioxidant properties of the plant, which intervenes in the neutralization of the free radicals of the oxidizing agents used [41,42]. The antioxidant action of flavonoids is well understood, flavonoids scavenging free radicals through chelation of trace metal ions, iron chelation, and oxidative stress reduction [30,43,44].

In comparison with other species of *Cynara* spp., our results show that the leaves of the *C. humilis* plant have a higher DPPH scavenging capacity than the leaves of *C. cardunculus*, which showed a value of 50 μg/mL [34]. The following compounds were found in the phytochemical analysis of the *C. humilis* plant: chlorogenic acid, caffeic acid, isochlorogenic acid, quercetin, luteolin, maritimen, apigenin [27]. Quercetin is a powerful antioxidant with a high scavenging capacity for DPPH free radicals [45]. Apigenin and luteolin, as well as their glycosides, have been seen to be potent antioxidants in other research [46]. The antioxidant activities of luteolin and luteolin 7-glucoside, have previously been reported against DPPH free radical scavenging [46,47]. 

Phenolic acids are known for their antioxidant effect, including chlorogenic acid. These antioxidant properties are almost the same as those o 4-O-caffeoylquinic acid and 3-O-caffeolquinicf when analyzing the radical scavenging of superoxide anionic radical and ability to inhibit of the oxidation of methyl linoleate [48].

### 2.4. Antimicrobial Activity

#### Disc Inhibitory Assay

To study their antimicrobial action, five bacterial strains were tested using a solid medium disc. The inhibitory zone (DIZ) was between 12 and 15 mm in diameter (Table 4). The highest inhibition zone was recorded in two strains, *E. coli* CIP 53126 and coagulase-negative *Staphylococcus*, in the root extract of 15.00 mm, followed by *C. albicans* ATCC 10231 which recorded a DIZ of 14.67 ± 1.15 mm in the root extract and 14.00 mm in the leaf extract. The lowest DIZ was recorded in the two strains *E. coli* ATCC 25922 and *K. pneumoniae* susceptible in the root extract of 12.00 mm. Statistical analysis of the inhibition zone diameter values of the two *C. humilis* extracts was significant (*p <* 0.05) against coagulase-negative *Staphylococcus* and *E. coli* CIP 53126, and not significant (*p* > 0.05) against the three remaining strains *C. albicans ATCC 10231, K. pneumoniae*, and E. *coli* ATCC 25922. Compared to the standard antibiotics used streptomycin, fluconazole and ampicillin, Coagulase negative *Staphylococcus* recorded a DIZ of 9.61 ± 0.20 mm in the presence of streptomycin, while the other bacterial strains showed resistance to the streptomycin and ampicillin. For *C. albicans* recorded a zone of inhibition of 21.20 ± 4.20 mm in the presence of fluconazole. In comparison with other species of *Cynara* spp., *Salmonella typhimurium, Bacillus subtilis, S. epidermidis,* and *E. coli* were all found to have antibacterial properties in *C. cardunculus.* On the other hand, *C. scolymus* showed antimicrobial effect against *S. aureus, B. subtilis, E. coli*, *Agrobacterium tumefaciens, Micrococcus luteus, Pseudomonas aeruginosa*, and *C. albicans* [46,49].

The extracts minimum inhibitory concentration (MIC) values were assessed against all of the microorganisms that were tested. Based on the finding in Table 5, the MIC recorded in this study was between 6.25 and 20 mg/mL. the MIC value needed to stop coagulase-negative *Staphylococcus* from growing was 6.25 mg/mL in the leaf extract, and 12.5 mg/mL in the root extract. *E. coli* CIP 53126 and E. coli ATCC 25922 were arrested at minimum concentration of 12.5 mg/mL in the root extract and 20 mg/mL in the leaf extract. At a minimum dose of 20 mg/mL, the other strains *K. pneumoniae* and *C. albicans* were suppressed.

Previous studies have shown that the highest antimicrobial effect could be related to the phenolic chemicals, in particular the presence of structural hydroxyl-phenolic groups, which increases the antimicrobial impact [50]. Further studies have confirmed the antimicrobial activity of polyphenols and flavonoids [51]. Indeed, the actuation mechanism of polyphenols in the antimicrobial activity is carried out by modifying the structure of the cell membrane, regulating its permeability, regulating cellular interactions by creating hydrogen bonds, and also decreasing the lipid content and finally preventing microbial development [52,53,54,55,56]. *C. humilis* is rich in polyphenols, especially chlorogenic acid, caffeic acid, isochlorogenic acid; in this sense, the antimicrobial effect of chlorogenic acid and caffeic acid has been well reported in the literature [6,49]

Luteolin, luteolin 7-rutinoside, and chlorogenic acid are components of *C. humilis.* Studies have shown compounds isolated from *C. cardunculus.* Among them, luteolin, luteolin 7-rutinoside, and chlorogenic acid exhibited more potent antimicrobial activities, and with an MIC varying between 0.03 and 0.10 mg/mL, the most active compound has been luteolin [46]. In another study, components extracted from the leaves of *C. scolymus*, among them luteolin 7-rutinoside and chlorogenic acid, were found to have a potential antibacterial action, with a MICs varying from 0.05–0.20 and 0.03–0.10 mg/mL [49]. Previous research has said that it has antibacterial properties of luteolin, apigenin, and other flavones [57,58]

### 2.5. Correlation between the Quality Parameters of C. humilis Extracts That Were Studied

According to the results displayed in Table 6, many of the factors examined in this study demonstrated a positive link between several of them. The IC_50_ shows a substantial positive correlation with *E. coli* CIP 53126 and *Staphylococcus* strains (r = 0.960 and r = 0.890, respectively), whereas the TFC has a link correlation with *E. coli* ATCC 25922 strain (r = 0.700), and the TPC with TAC (r = 0.626). With *E. coli* ATCC 25922 strains, the TAC exhibits a positive association (r = 0.750). A high correlation exists between *E. coli* CIP 53126 and *Staphylococcus* (r = 0.891). The TAC, on the other hand, has a strong negative association with *Staphylococcus* (r = −0.925), and the TPC in turn has a strong negative correlation with IC_50_ and *Staphylococcus* (r = −0.836 and r = −0.710, respectively). Additionally, a negative correlation was found for the TCF with *C. albicans* ATCC 10231 (r = −0.765) and the *E. coli* ATCC 25922 with *Staphylococcus* and *E. coli* CIP 53126 strains (r = −0.791 and r = −0.777, respectively). 

Overall, the considerable the link between polyphenols and flavonoids and free radical scavenging indicates that *C. humilis* possesses potent antioxidant properties. Our results confirm previous studies that polyphenols and flavonoids have a strong free radical scavenging capacity and exert antioxidant activity [39,59,60,61].

## 3. Materials and Methods 

### 3.1. Reagents and Apparatus 

The following compounds, Na_2_CO_3_, AlCl_3_, Folin-Ciocalteu (RE00180250), potassium acetate, quercetin, sulfuric acid 0.6 M; sodium phosphate 28 mM, 4 mM ammonium molybdate, methanol, ethanol, 2,2-diphenyl-1-picrylhydrazyl, and NaCl were provided to Sigma-Aldrich, St. Louis, MO, États-Unis. Spectrophotometry (SOMESTIM, Rabat, Morocco) to determine the absorption capacity. 

### 3.2. The Plant Used and Extraction Process

The study area is the region of Moulay Yacoub, the region is located 25 km northwest of the city fez and its area is estimated at 1700 km^2^. the climate is characterized by high precipitation during the winter season, the annual average is estimated at 600 mm/year. For temperature, it is estimated between 6 to 36 °C, with a very hot summer. The region consists of a sedimentary series dominated by marly deposits [1,62]. The annual herb *C. humilis* pertains to the Asteraceae family (Figure 2), easily recognizable by its finely dissected leaves and its 4-sided cypselas with wing-like veins ending in a rigid yellowish spine. Unbranched taproot. Stem: has two rings of vascular bundles. Receptacle: slightly concave to slightly convex. Period of flowering: mainly from May to July. The plant is native to: Algeria, Canary Islands, Morocco, Portugal, and Spain [21,23]. The leaves and roots of *C. humilis*, which were employed in this study, were gathered in the Moulay Yacoub area, Morocco, from March to April 2020. The plant was identified (specimen number is MY002) by the botanist Lahcen Zidane, professor at the Faculty of Science of Kenitra, University Ibn Tofail, Kenitra, Morocco. Water was used to clean the plant’s parts, dried at 35 °C, and powdered using a mortar. An amount of 50 g of dry powder plant weight was used for extraction, and the dry extract was obtained by maceration using a mixture of 70% ethanol and 30% distilled water, which means 10% of the plant powder (mass/volume). The macerate was filtered, then a rotary evaporator at 40 °C under partial vacuum was used to evaporate the solvent, finally, the acquired dry extract was utilized.

The extraction yield of *C. humilis* roots and leaves was determined using the following report:Y (%) = (ME/MP) × 100
Y: Yield of extraction (%); MP: Mass of plant powder used (g); ME: Mass of the crude extract obtained (g). 

### 3.3. Determination of the Phenolic Content (TPC)

The Folin-Ciocalteu technique was used to estimate the phenolic compounds in the plant with modest alterations [30]. Plant extracts were mixed with 4 mL of 2 percent Na_2_CO_3_ and 200 μL of Folin-Ciocalteu reagent in this experiment, the absorbance has been determined spectrophotometrically at 670 nm after the solution had been left for thirty minutes. The equation is used to calculate the phenolic composition of the plant extracts was established from the calibration range of gallic acid, y = 5.4067x + 0.2213 and *R*^2^ = 0.9304.

### 3.4. Determination of the Flavonoid Content (TFC)

The following procedure has been used to investigate the flavonoids composition [53]. An amount of 0.2 mL of AlCl_3_ 10% and 2 mL of the plant extracts has been blended with 7.6 mL of ethanol and 0.2 mL of 1 M potassium acetate. After leaving the solution for forty min, at 430 nm, the absorbance has been measured spectrophotometrically. The equation used to calculate the content of flavonoids was established from quercetin’s calibration curve, y = 0.0062x + 0.4289 and *R*^2^ = 0.857. 

### 3.5. Antioxidant Assay 

#### 3.5.1. Total Antioxidant Capacity (TAC)

To study the antioxidant assay TAC the following method was employed after a modest adjustment [63]. In this assay, 50 μL of the plant extract was combined with 2 mL of a solution prepared from sulfuric acid 0.6 M, sodium phosphate 28 mM, and 4 mM ammonium molybdate. Afterwards, the reaction mixture has been placed in a water bath at 95 °C for thirty minutes. A spectrophotometer set at 695 nm has been used to measure the absorbance of the reaction medium, with a 50 μL tube of methanol without extract serving as a negative control. A calibration curve utilizing ascorbic acid has been used to asses the total antioxidant capacity of the extract, the finding TAC values has been written by the unit (mg EAA/g of extract), which means milligrams of ascorbic acid equivalent by gram of extract. 

#### 3.5.2. Scavenging of the Free Radical DPPH

The following method [64] was employed to evaluate the DPPH antioxidant activity. In this assay, 0.75 mL of 2,2-diphenyl-1-picrylhydrazyl (0.004) has been mixed with 0.2 mL of each dilution series of the extracts studied. After leaving the solution for thirty minutes in the dark, the spectrophotometer has been determined at 517 nm.

The percentage of inhibition (PI) has been determined utilizing the report bellow: PI (%) = ((A_0_ − A)/A_0_) × 100

A_0_: the absorbance value in the solution (2,2-diphenyl-1-picrylhydrazyl) which does not contain the extract. 

A: the absorbance value in the solution (2,2-diphenyl-1-picrylhydrazyl) containing the extract.

### 3.6. Antimicrobial Activity of C. humilis

To study the antimicrobial test five bacterial strains were employed including two strains of *Escherichia coli*: *E. coli* CIP 53126, *E. coli* ATCC 25922, coagulase-negative *Staphylococcus*, *K. pneumoniae* susceptible, and a single fungal strain of *C. albicans* ATCC 10231. The microbial strains utilized in this assay were taken from the FMP-Fez microbiology laboratory, Morroco. Colonies were collected employing 24 h cultures to create bacterial suspensions, and the cultures were kept at 4 °C to preserve the cultures prepared in agar MH. The colonies that had been prepared were suspended and shook for 15 s in sterile solution (0.9% NaCL). The density was set to 0.5 turbidity (equivalent to 1–5 × 10^8^ CFU/mL) [65].

#### 3.6.1. Agar Disc Diffusion

The following methodology was used to examine the agar disk diffusion assay [64]. Bacterial suspensions were used to prepare MH agar plates (10^8^ CFU/mL), then inoculated by swabbing. After, Whatman paper discs (6 mm) were laid on the surface of the pre-inoculated agar. Then, a volume of 20 μL of 50 mg/mL of *C. humilis* was used to dip each Whatman paper disc. A disc of Whatman tempered in dimethyl sulfoxide was used as a negative control in the medium of each Petri dish, the plates were then incubated at 37 °C for 24 h, and the widths of the inhibitory zone (DIZ) were calculated. The bacteria have been categorized to use the latter (DIZ) [20].

○Non-sensitive: when the value of (DIZ) did not exceed 8 mm.○Sensitive: when the value of (DIZ) was found in the interval 9 to 14 mm.○Very sensitive: when the value of (DIZ) was found in the interval 15 to 19 mm.○Extremely sensitive: when the value of (DIZ) exceeded 20 mm. 

#### 3.6.2. Minimum Inhibitory Concentration (MIC)

To investigate the test for the minimal inhibitory concentration, 96-well plates were employed in accordance with NCCLS guidelines [66], with minor adjustments. Using dimethyl sulfoxide, extracts from the leaves and roots of *C. humilis* were synthesized in sterile hemolysis tubes. The concentration of the extracts in the wells was prepared from successive 1:1 dilution in Mueller Hinton broth mixture. Arriving at values in the range of 0.039 to 20 mg/mL following that, 50 μL of microbial suspensions were added to 5 μL of Mueller Hinton broth, with various concentrations of the were applied to estimate MIC value. After that, the plates were kept at 37 °C for 18 h, after which 40 μL of 0.5% triphenyl tetrazolium chloride was added to each well. The MIC value has been calculated utilizing the smallest concentration that won’t result a red color [65].

### 3.7. Analytical Statistics

The means and standard deviations were used to present the results of this study’s findings. The one-way ANOVA and student’s test used to evaluate the statistical difference. Tukey’s test is often used as a post hoc test to analyze the significant difference in TAC and DPPH assay, Minitab 19.1 is the software used in this work, *p <* 005 was utilized to designate the significant difference in all analyzed tests. 

## 4. Conclusions

This study determined the chemical constitution and antioxidant and antimicrobial biological properties of the extracts of the roots and leaves of the plant *C. humilis* collected in spring 2020 in one locality in Morocco. The results indicate that leaves are richer in polyphenols (38.84 mg GAE/g extract) and flavonoids (27.07 mg QE/g extract) than the roots 29.56 mg GAE/g extract and 24.39 mg QE/g extract, respectively). The TAC antioxidant assay showed that the leaf extract (0.77 mg AAE/g extract) has higher antioxidant power than roots (0.60 mg AAE/g extract). In the DPPH antioxidant assay, the IC_50_ values in the leaf and root extract were 0.23 and 0.93 µg/mL, respectively. These extracts demonstrated antimicrobial effect on all strains studied, approved by a zone of inhibition with a diameter varying from 12 to 15 mm, and an MIC value of 6.25 mg/mL against coagulase-negative *Staphylococcus* in the root extract. These findings can be used in industry of pharmaceuticals to take care of microbial disorders and create medications to combat resistant bacteria, as well as in the food industry to preserve food using natural preservations rather than synthetic preservatives that have numerous adverse effects. Given that this is the first study on this plant, and only from one locality in Morocco, it is also difficult to judge the overall polyphenol content; therefore, additional research and repeated analyses in at least spring, summer, and autumn, during at least two growing seasons, and at several sites are needed to confirm these findings and identify the substances responsible for these biological activities.

## Figures and Tables

**Figure 1 plants-11-01200-f001:**
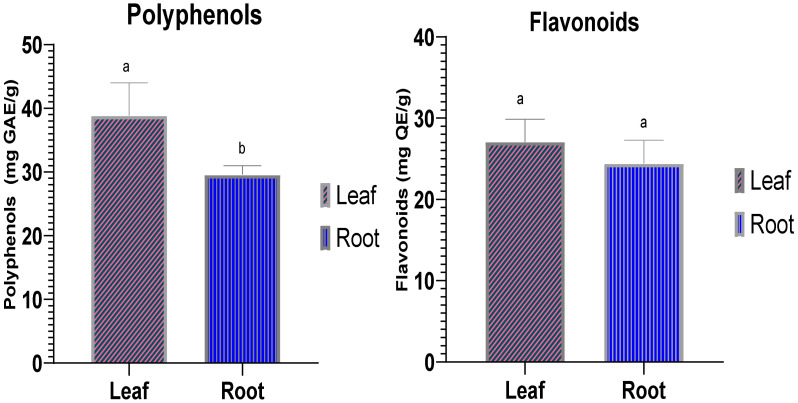
Histogram of flavonoid contents (TFC) and polyphenol content (TPC) in the *C. humilis* extracts. The significant difference (*p* < 0.05) between the tested samples is shown by the two tetters a and b.

**Figure 2 plants-11-01200-f002:**
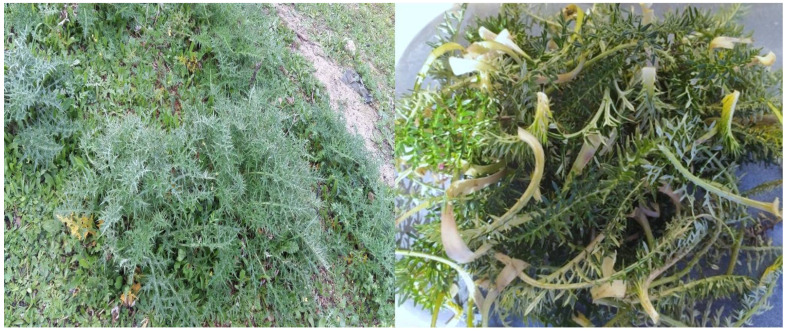
The plant studied, *Cynara humilis*.

**Table 1 plants-11-01200-t001:** Yield obtained from the extraction of roots and leaves of *C. humilis*.

Extract	Mass of Plant Powder Used (g)	Mass of the Crude Extract Obtained (g)	Yield of Extraction (%)
Leaf	50	8.7	17.4
Root	50	6.4	12.8

**Table 2 plants-11-01200-t002:** The raw data of the analyses carried out on the *C. humilis* plant with three repetitions.

Extract	TPC	TFC	IC_50_	TAC	*E. coli* ATCC25922	*E. coli* CIP 53126	*K. pneumoniae*	*Staphylococcus*	*C. albicans*
Leaf	32.93	26.74	0.23	0.77	13	12	12	12	14
Leaf	42.33	30.01	0.21	0.84	13	13	12	12	14
Leaf	41.27	24.47	0.25	0.7	12	13	13	13	14
Root	27.94	26.66	0.94	0.63	12	15	12	15	14
Root	30.03	25.34	0.92	0.55	12	15	12	16	14
Root	30.71	21.16	0.95	0.62	12	15	12	14	16

**Table 3 plants-11-01200-t003:** DPPH antioxidant test result represented by IC_50_.

Extract	IC_50_ (μg/mL)
Leaf	0.23 ^B^ ± 0.02
Root	0.93 ^A^ ± 0.01
BHT	0.20 ^C^ ± 0.00
Quercetin	0.05 ^E^ ± 0.00
Ascorbic acid	0.16 ^D^ ± 0.01

The letters ^A–E^ indicate that there is a statistical difference (*p* < 0.05) between the tested samples.

**Table 4 plants-11-01200-t004:** DIZ values of the two extracts of *C. humilis*.

Strains of Bacteria	The Zones of Inhibition’s Diameter (mm)				
	Leaf Extract	Root Extract	Ampicillin	Streptomycin	Fluconazole
*E. coli* CIP 53126	12.67 ^B^ ± 0.57	15.00 ^A^ ± 0.00	R	R	---
E. coli ATCC 25922	12.67 ^A^ ± 0.57	12.00 ^A^ ± 0.00	R	R	---
Coagulase-negative *Staphylococcus*	12.33 ^B^ ± 0.57	15.00 ^A^ ± 0.00	R	9.61 ± 0.20	---
*K. pneumoniae* sensible	12.33 ^A^ ± 0.57	12.00 ^A^ ± 1.00	R	R	---
*C. albicans* ATCC 10231	14.00 ^A^ ± 0.00	14.67 ^A^ ± 1.15	---	---	21.20 ± 4.20

ATCC: American Type Culture Collection; CIP: Institut Pasteur Collection; R: resistant; ---: this antibiotic is ineffective against this strain; the difference in the two extracts is significant statistically (*p* < 0.05), as seen in ^A,B^.

**Table 5 plants-11-01200-t005:** *C. humilis* extracts and their minimum inhibitory concentration (MIC) values.

Strains of Bacteria	Concentration (mg/mL)	
	Leaf Extract	Root Extract
*E. coli* CIP 53126	20	12.5
*E. coli* ATCC 25922	20	12.5
Coagulase-negative *Staphylococcus*	6.25	12.5
*K. pneumoniae* sensible	20	20
*C. albicans* ATCC 10231	20	20

**Table 6 plants-11-01200-t006:** Pearson coefficients of correlation between *C. humilis* parameters.

Studied Parameters	TPC	TFC	IC_50_	TAC	*E. coli* ATCC 25922	*E. coli* CIP53126	*K. pneumoniae*	*Staph*
TFC	0.407							
IC_50_	−0.836	−0.531						
TAC	0.626	0.318	−0.688					
*E. coli* ATCC 25922	0.435	0.700	−0.725	0.750				
*E. coli* CIP 53126	−0.655	−0.466	0.960	−0.663	−0.777			
*K. pneumoniae*	0.567	−0.211	−0.424	−0.024	−0.316	−0.307		
*Staph*	0.710	−0.384	0.890	−0.925	−0.791	0.891	−0.200	
*C. albicans*	0.280	−0.765	0.464	0.166	−0.316	0.430	−0.200	0.100

## Data Availability

Not applicable.

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
