# Peer review of "In Vitro Studies on the Antimicrobial and Antioxidant Activities of Total Polyphenol Content of *Cynara humilis* from Moulay Yacoub Area (Morocco)"

_plants, 2022, doi:10.3390/plants11091200_

Round 1

Reviewer 1 Report

Dear authors, dear Editor, 

the second version of this paper is much, much better written, congratulations!

Author Response

Thank you for reviewing our manuscript and for the reviewers’ comments concerning our paper Ref: molecules-1686788. we are glad to have received your favourable comments. In view of the encouraging comments, we have thoroughly revised the manuscript. 

Reviewer 2 Report

Review comments to the author

Title: ''In vitro Studies on the Antimicrobial and Antioxidant Activities of Total Polyphenols of spring content of Cynara humilis from Moulay Yacoub area (Morocco)''.

Manuscript ID: plants-1686788.

Abstract:

1- Page 1, Lines 21-22: Please, re-write this sentence ''Abstract: Abstract: In Morocco, Cynara humilis L. has been made use of traditionally used as a 21 medicinal plant'', and delete the repeated word ''Abstract''.

2- Page 1, Lines 22-23: The sentence ''The objective of this research is to study the antimicrobial and antioxidant biological properties of....'' should be modified to ''The objective of this research is to study the antimicrobial and antioxidant activities of....''.

3- Page 1, Lines 32: The section ''....in root extracts'' should be modified to ''....in root extract''.

Results and Discussion

1- Page 3, Line 99: The word ''several'' should be typed as ''Several''.

2- Page 3, Lines 103-104: The sentence ''HPLC chromatographic analyses have shown that the C. humilis plant is rich in polyphenols : caffeic acid, chlorogenic acid, isochlorogenic acid [27].'' should be typed as ''HPLC chromatographic analyses have shown that the C. humilis  plant is rich in polyphenols such as caffeic acid, chlorogenic acid and isochlorogenic acid [27].''

3- Page 4, Line 125: Table (1) should be moved after text to the line 160.

4- Table (2) should be mentioned in the text.

5- Page 5, Lines 166-181: All names of microbial strains should be typed in the same way; complete names or on abbreviated names, and in all coming positions  

6- Antimicrobial activities should be compared with standard antibiotics.  

  1. Materials and methods

3.1. Reagents and apparatus

1- Page 7, Lines 232: In the molecular formula ''Na2CO3, AlCl3'' the numbers should be typed in subscript fonts.

3.2. Plant Material and Extraction Process

1- A voucher specimen number should be mentioned.

2- The weight of dry powdered plant used in extraction should be mentioned.

 2- The extraction yield or total extractable content (TEC) should be mentioned.

  1. Conclusion

1- Page 9, Lines 341: In the term ''IC50'' the number ''50'' should be typed in subscript font.

Abbreviations:

- List of abbreviations should be inserted by the end of the manuscript before references.

Author Response

Title: ''In vitro Studies on the Antimicrobial and Antioxidant Activities of Total Polyphenols of spring content of Cynara humilis from Moulay Yacoub area (Morocco)''.

Thank you for reviewing our manuscript and for the reviewers’ comments concerning our paper Ref: molecules-1686788.Those comments are all valuable and very helpful for revising and improving our paper, as well as the important guiding significance to our researches. In view of the encouraging comments, we have thoroughly revised the manuscript.

The point wise responses for the reviewer:

Manuscript ID: plants-1686788.

Abstract:

  • Page 1, Lines 21-22: Please, re-write this sentence ''Abstract: Abstract: In Morocco, Cynara humilis  has been made use of traditionally used as a 21 medicinal plant'', and delete the repeated word ''Abstract''.

Response: Thank you very much for your suggestion. We have taken your suggestion in consideration. The sentence is re-write and  highlighted in blue. Line 21.

  • Page 1, Lines 22-23: The sentence ''The objective of this research is to study the antimicrobial and antioxidant biological properties of....'' should be modified to ''The objective of this research is to study the antimicrobial and antioxidant activities of....''.

Response: We have taken your suggestion in consideration. The sentence is modified. Line 22-23.

3- Page 1, Lines 32: The section ''....in root extracts'' should be modified to ''....in root extract''.

Response: We have taken your suggestion in consideration. Line 31-32

Results and Discussion

  • Page 3, Line 99:The word ''several'' should be typed as ''Several''.

Response: it was done. Line 102.

  • Page 3, Lines 103-104: The sentence ''HPLC chromatographic analyses have shown that the  humilisplant is rich in polyphenols : caffeic acid, chlorogenic acid, isochlorogenic acid [27].'' should be typed as ''HPLC chromatographic analyses have shown that the C. humilis  plant is rich in polyphenols such as caffeic acid, chlorogenic acid and isochlorogenic acid [27].''

Response: We have taken your suggestion in consideration. The sentence has been rephrased. Line 106-108.

  • Page 4, Line 125: Table (1) should be moved after text to the line 160.

Response: We have taken your suggestion in consideration. Table (1) has been moved after text to the line 165.

  • Table (2) should be mentioned in the text.

Response: We have taken your suggestion in consideration. Table (2) has been mentioned in the line 130.

  • Page 5, Lines 166-181: All names of microbial strains should be typed in the same way; complete names or on abbreviated names, and in all coming positions.

Response: We have taken your suggestion in consideration. Table (2) has been mentioned in the line 130.

6- Antimicrobial activities should be compared with standard antibiotics.  

 Response: We have taken your suggestion in consideration. The comparison was added in the table 4 and the mentioned in the text. Line 190-194.

  1. Materials and methods

3.1. Reagents and apparatus

1- Page 7, Lines 232: In the molecular formula ''Na2CO3, AlCl3'' the numbers should be typed in subscript fonts.

 Response: We have taken your suggestion in consideration. it has been set. Line 256.

3.2. Plant Material and Extraction Process

1- A voucher specimen number should be mentioned.

Response: A voucher specimen number was mentioned in the text. Lines 270-271.

  • The weight of dry powdered plant used in extraction should be mentioned.

Response: The weight of dry powdered plant used in extraction was mentioned in the text. Lines 273-274.

 2- The extraction yield or total extractable content (TEC) should be mentioned.

Response: The extraction yield or total extractable content (TEC) was mentioned in the text. Lines 278-284.

  1. Conclusion
  • Page 9, Lines 341: In the term ''IC50'' the number ''50'' should be typed in subscript font.

Response: it was done. Line 370.

Abbreviations:

- List of abbreviations should be inserted by the end of the manuscript before references.

Response: Thank you very much for your suggestion. we removed it during the previous revision.

According to MDPI Instructions for Authors, And,

According to the comments of the reviewer during the last revision before the resubmission.  “Regarding Abbreviations, that part must be removed. Please check  Instructions for authors https://www.mdpi.com/journal/plants/instructions and apply them as it is mentioned: "Acronyms/Abbreviations/Initialisms should be defined the first time they appear in each of three sections: the abstract; the main text; the first figure or table. When defined for the first time, the acronym/abbreviation/initialism should be added in parentheses after the written-out form". Instructions for Authors are given to be respected, they are not optionally.

However, if you find that it is necessary to add it, we will do it in the proofreading manuscript.

Reviewer 3 Report

This is a resubmitted manuscript. Please see my suggestions:

L90. After reference [30-32], the sentence must be completed as follows: the characteristic structure of each of these afore mentioned compounds indicating a specific behaviour / action / role, as it was statistically demonstrated [Glevitzky I., et al. Statistical Analysis of the Relationship Between Antioxidant Activity and the Structure of Flavonoid Compounds. Rev. Chim. 2019, 70(9), 3103-3107. https://doi.org/10.37358/RC.19.9.7497]

L116. The sentence must be removed. It is repetitive with L77-78.

Text mentioning that a Table or a figure will be inserted must be BEFORE that Table/Figure not after.

  • Table 1 is before mentioning it in L133. Please insert the Table after that paragraph. Also auto-fit the content for Table 1, the table is much too large.
  • Table 2 is not mentioned at all in the main text. Please complete the main text before inserting the table. Moreover, in the head of the table, please complete the first cell (it is not allowed empty cell in a Table) as "Part of the plant".
  • Table 4 must be moved after mentioning it in L186.

Abbrev. CIP and ATCC must be explained under Tables 3 and 4.

First cell of Table 5, in the head of the table, needs name/title

Author Response

Thank you for reviewing our manuscript and for the reviewers’ comments concerning our paper Ref: molecules-1686788.Those comments are all valuable and very helpful for revising and improving our paper, as well as the important guiding significance to our researches. In view of the encouraging comments, we have thoroughly revised the manuscript.

The point wise responses for the reviewer:

This is a resubmitted manuscript. Please see my suggestions:

L90. After reference [30-32], the sentence must be completed as follows: the characteristic structure of each of these afore mentioned compounds indicating a specific behaviour / action / role, as it was statistically demonstrated [Glevitzky I., et al. Statistical Analysis of the Relationship Between Antioxidant Activity and the Structure of Flavonoid Compounds. Rev. Chim. 2019, 70(9), 3103-3107. https://doi.org/10.37358/RC.19.9.7497]

Response: Thank you very much for your suggestion. We have taken your suggestion in consideration. The sentence completed and highlighted in blue. Lines 93 and 94.

L116. The sentence must be removed. It is repetitive with L77-78.

Response: We have taken your suggestion in consideration. The sentence is removed.

Text mentioning that a Table or a figure will be inserted must be BEFORE that Table/Figure not after.

  • Table 1 is before mentioning it in L133. Please insert the Table after that paragraph. Also auto-fit the content for Table 1, the table is much too large.

Response: We have taken your suggestion in consideration. The table and the auto-fit the content were inserted.

  • Table 2 is not mentioned at all in the main text. Please complete the main text before inserting the table. Moreover, in the head of the table, please complete the first cell (it is not allowed empty cell in a Table) as "Part of the plant".

Response: We have taken your suggestion in consideration. All suggestions were done.

  • Table 4 must be moved after mentioning it in L186.

Response: the table was moved.

Abbrev. CIP and ATCC must be explained under Tables 3 and 4.

Response: the abbreviation CIP and ATCC were added under the table 4 and 5. Lines 202 and 214.

First cell of Table 5, in the head of the table, needs name/title.

Response: it was added. 

This manuscript is a resubmission of an earlier submission. The following is a list of the peer review reports and author responses from that submission.

Round 1

Reviewer 1 Report

Respected authors,

You say that these are the first studies on antimicrobial and antioxidant activities of the species Cynara humilis.

There is no description of the Morocco area (climatic, pedological, orographic). The botanical description of the species and its distribution are missing. Do you have a (digitized) herbarium specimen stored in the official herbarium of plant material you analyzed? Habitat photos?

Are you sure this is not a local subspecies?

According to KEW C. humilis is native to: Algeria, Canary Is., Morocco, Portugal, and Spain.

It's important appropriate decribe locality and ecological conditions of collected plant material as well as to have herbarium specimen and photos.

It is very important to mention information, that you did not give in the text,

how many analyzes did you do for each parameter?

This is very important to evaluate security of assessed standard deviation and signification of your data. Please can you give the raw data in apendix?

That's why I will propose supplement to the title:

In vitro Studies on the Antimicrobial and Antioxidant Activities of Total Polyphenols of Cynara humilis L. of spring content of Cynara humilis from Moulay Yacoub area (Morocco).

It is also questionable to judge the overall composition of polyphenols, and you only researched in the spring part of the year at one site. It's nedded to emphasize this in the results and conclusion.

For more concrete conclusions, you lack more localities and repeated analyzes at least through spring, summer and autumn through at least two vegetation seasons.

Beside, first time when you mention the species, its needed to write the authors, later there is no need to repaet and it's common to use short of the name C. humilis. 

There is a lot of mistakes of italic- non italic (in text and literature), I suposse that's the mistake of technical part od editorial board, but it must be corrected.

Author Response

Response to Reviews and editors

We are thankful to the learner referee for reviewing our manuscript and suggested the valuable points for the betterment of the current work. We have considered all your questions and comments carefully and we have outlined each change made in the revised manuscript. The point responses to your comments and questions are presented below.

Response to Reviewer #1:

You say that these are the first studies on antimicrobial and antioxidant activities of the species Cynara humilis.

  • There is no description of the Morocco area (climatic, pedological, orographic). The botanical description of the species and its distribution are missing. Do you have a (digitized) herbarium specimen stored in the official herbarium of plant material you analyzed? Habitat photos?

Response: Thank you very much for your suggestion. We have taken your suggestion in consideration the added paragraph is highlighted in yellow. Line 199-208

  • Are you sure this is not a local subspecies?

Response: Thank you very much for your notice. Indeed, the plant was identified by the botanist Lahcen Zidane, professor at the Faculty of Science of Kenitra, University Ibn Tofail, Kenitra, Morocco. Line 210-212.

  • According to KEW  humilisis native to: Algeria, Canary Is., Morocco, Portugal, and Spain.

Response: Thank you very much for your suggestion. The added sentence is highlighted in yellow. Line 207-208.

  • It's important appropriate decribe locality and ecological conditions of collected plant material as well as to have herbarium specimen and photos.

Response: Thank you very much for your suggestion. We have taken your suggestion in consideration the added paragraph is highlighted in yellow. Line 199-208 and 217.

  • It is very important to mention information, that you did not give in the text,

Response: Thanks very much for your observation. We have taken your suggestion in consideration.

  • How many analyzes did you do for each parameter?

Response:  The number of analyses performed for each parameter is three. Line 485 and 489.

  • This is very important to evaluate security of assessed standard deviation and signification of your data. Please can you give the raw data in appendix?

Response:  We have taken your suggestion in consideration. The raw results have been added to the table in the appendix. Line 485 and 489.

  • That's why I will propose supplement to the title:
  • In vitro Studies on the Antimicrobial and Antioxidant Activities of Total Polyphenols of Cynara humilis  of spring content of Cynara humilis from Moulay Yacoub area (Morocco).

Response:  Thank you very much for your suggestion. We have rephrased the title. The added sentence is highlighted in yellow. Line 1-4

  • It is also questionable to judge the overall composition of polyphenols, and you only researched in the spring part of the year at one site. It's nedded to emphasize this in the results and conclusion.For more concrete conclusions, you lack more localities and repeated analyzes at least through spring, summer and autumn through at least two vegetation seasons.

Response:  Thanks very much for your observation. We have taken your suggestion in consideration. The added sentences are highlighted in yellow. Line 293-296.

  • Beside, first time when you mention the species, its needed to write the authors, later there is no need to repeat and it is common to use short of the name  humilis. 

Response: Thanks very much for your remark.  It is corrected all along the manuscript

  • There is a lot of mistakes of italic- non italic (in text and literature), I suposse that's the mistake of technical part od editorial board, but it must be corrected.

Response: Thanks very much for your notice. All corrections have been made all along the manuscript.

Reviewer 2 Report

Review comments to the author

Title: ''In vitro Studies on the Antimicrobial and Antioxidant Activities of Total Polyphenols of Cynara humilis L. ''.

Manuscript ID: plants-1649548.

Abstract:

1- Page 1, Lines 22-24: The section ''The content of poly-phenols and flavonoids was evaluated with the help of reagents Folin-Ciocalteu and aluminum chloride'' could be modified to be ''The content of poly- phenols and flavonoids was evaluated using Folin-Ciocalteu's and aluminum  chloride assays''.

2- Page 1, Line 25: The section ''In antimicrobial assays, five strains were.......'' could be modified to be ''In antimicrobial assays, five pathogenic microbial strains were.......''.

3- Page 1, Lines 26-27: The scientific names of the pathogenic microbial strains (e.g., Escherichia coli, Staphylococcus, Klebsiella pneumoniae, and Candida albicans) should be typed in italic font here and in all coming positions through MS.

4- Type of extract should be mentioned.

5- Page 1, Line 31: The term ''IC50'' should be typed in proper form (There is no any space between the number 50 and IC letters, also the number 50 should be typed in subscript font).

6- Page 1, Lines 32-33: The section ''C. humilis extracts exhibited an inhibitory effect of the five strains examined. With inhibition zones ranging from 12.00 to 15.00 mm'' could be modified to be ''C. humilis extracts exhibited an inhibitory effect against five tested strains with inhibition zones ranging from 12.00 to 15.00 mm''.

7- Page 1, Lines 33-35: The section ''In the presence of the root extract, the lowest minimum inhibitory concentration against coagulase-negative Staphylococcus was 6.25 mg/mL'' could be modified to be '' Moreover, the root extract showed the lowest minimum inhibitory concentration against coagulase-negative Staphylococcus with IC50 value of 6.25 mg/mL''.

8- Page 1, Line 36: The plant name ''C. humilis'' should be typed in italic font.

  1. Results:

1- The title ''Results'' should be replaced by ''Results and Discussion''.

2- Page 2, Line 76: The plant name ''Cynara humilis'' should be typed as ''C. humilis''.

3- Page 2, Line 78: In the term ''R2'' the number 2 should be typed in superscript font.

4- Page 3, Line 114: The sub-title ''2.2.2. Scavenging of the Free Radical DPPH.'' should be replaced by ''2.2.2. DPPH Free Radical Scavenging Antioxidant Activity''.

5- Page 3, Line 122: The plant name ''C. humilis'' should be typed in italic font.

6- Page 4, Lines 127-128: The plant names ''C. cardunculus and C. scolymus'' should be typed in italic fonts.

7- Page 4, Line 130: The citation [35] should be typed in non-italic fonts.

8- Page 4, Line 133: The plant name ''C. humilis'' should be typed in italic font.

9- Antimicrobial activity results should be compared with standard antibiotics.

  1. Materials and Methods:

3.1. Plant Material

1- Page 5, Line 192: The sub-title ''3.1. Plant Material'' should be typed as ''3.1. Plant Material and Extraction Process''.

2- The plant identifier should be mentioned.

3- A voucher specimen number should be mentioned.

4- If allowed, the tested extracts should be subjected to further phytochemical analysis like HPLC-DAD-ESI-MS/MS analysis followed by molecular modelling studies like docking.  

  1. Conclusion

1- Conclusion should be supported by the promising results.  

Abbreviations:

- List of abbreviations should be inserted by the end of the manuscript before references.

References:

1- Scientific names of plant names and microbial strains should be typed in italic font and in proper font.

Author Response

Response to Reviews and editors

We are thankful to the learner referee for reviewing our manuscript and suggested the valuable points for the betterment of the current work. We have considered all your questions and comments carefully and we have outlined each change made in the revised manuscript. The point responses to your comments and questions are presented below.

Response to Reviewer #2:

Abstract:

  • Page 1, Lines 22-24: The section ''The content of poly-phenols and flavonoids was evaluated with the help of reagents Folin-Ciocalteu and aluminum chloride'' could be modified to be ''The content of poly- phenols and flavonoids was evaluated using Folin-Ciocalteu's and aluminum  chloride assays''.

Response: Thank you very much for your suggestion. We have taken your suggestion in consideration the added sentence is highlighted in yellow. Line 23-24

  • 2- Page 1, Line 25: The section ''In antimicrobial assays, five strains were.......'' could be modified to be ''In antimicrobial assays, five pathogenic microbial strains were.......''.

Response: Thank you very much for your suggestion. We have taken your suggestion in consideration the added sentence is highlighted in yellow. Line 26-27

  • 3- Page 1, Lines 26-27: The scientific names of the pathogenic microbial strains (e.g., Escherichia coli, Staphylococcus, Klebsiella pneumoniae, and Candida albicans) should be typed in italic font here and in all coming positions through MS.

Response: Thanks very much for your observation. We have taken your suggestion in consideration. Line 27-28

  • Type of extract should be mentioned.

Response: We have taken your suggestion in consideration. Type of extract is mentioned in yellow. Line 23 and 37.

  • 5- Page 1, Line 31: The term ''IC50'' should be typed in proper form (There is no any space between the number 50 and IC letters, also the number 50 should be typed in subscript font).

Response: Thank you very much for your suggestion. We have taken your suggestion in consideration, the added IC50 is highlighted in yellow. Line 32 and 36.

  • 6- Page 1, Lines 32-33: The section ''C. humilis extracts exhibited an inhibitory effect of the five strains examined. With inhibition zones ranging from 12.00 to 15.00 mm'' could be modified to be ''C. humilis extracts exhibited an inhibitory effect against five tested strains with inhibition zones ranging from 12.00 to 15.00 mm''.

Response: Thank you very much for your suggestion. We have taken your suggestion in consideration. Line 33-34

  • 7- Page 1, Lines 33-35: The section ''In the presence of the root extract, the lowest minimum inhibitory concentration against coagulase-negative Staphylococcus was 6.25 mg/mL'' could be modified to be '' Moreover, the root extract showed the lowest minimum inhibitory concentration against coagulase-negative Staphylococcus with IC50 value of 6.25 mg/mL''.

Response: Thank you very much for your suggestion. We have taken your suggestion in consideration the added sentence is highlighted in yellow. Line 34-36.

  • 8- Page 1, Line 36: The plant name ''C. humilis'' should be typed in italic font.

Response: We have taken your suggestion in consideration. Line 37.

  1. Results:
  • The title ''Results'' should be replaced by ''Results and Discussion''.

Response: You are right. We have taken your suggestion in consideration. Line 81.

  • 2- Page 2, Line 76: The plant name ''Cynara humilis'' should be typed as ''C. humilis''.

Response: Thanks very much for your observation. We have taken your suggestion in consideration.

  • 3- Page 2, Line 78: In the term ''R2'' the number 2 should be typed in superscript font.

Response: Thanks very much for your observation. We have taken your suggestion in consideration.

  • 4- Page 3, Line 114: The sub-title ''2.2.2. Scavenging of the Free Radical DPPH.'' should be replaced by ''2.2.2. DPPH Free Radical Scavenging Antioxidant Activity''.

Response: Thanks very much for your observation. The sentence has been rephrased. Line 116

  • 5- Page 3, Line 122: The plant name ''C. humilis'' should be typed in italic font.

Response: Thanks very much for your observation. We have taken your suggestion in consideration.

  • 6- Page 4, Lines 127-128: The plant names ''C. cardunculus and C. scolymus'' should be typed in italic fonts.

Response: Thanks very much for your observation. We have taken your suggestion in consideration. Line 130-131.

  • 7- Page 4, Line 130: The citation [35] should be typed in non-italic fonts.

Response: Thanks very much for your observation. We have taken your suggestion in consideration. Line 122.

  • 8- Page 4, Line 133: The plant name ''C. humilis'' should be typed in italic font.

Response: Thanks very much for your observation. We have taken your suggestion in consideration. Line 125.

  • 9- Antimicrobial activity results should be compared with standard antibiotics.

Response: Thank you very much for your suggestion. We did not perform the standard antibiotic test. The antimicrobial activity was compared between the extracts studied.

  1. Materials and Methods

3.1. Plant Material

  • Page 5, Line 192: The sub-title ''3.1. Plant Material'' should be typed as ''3.1. Plant Material and Extraction Process''.

Response: Thank you very much for your suggestion. We have taken your suggestion in consideration, the added sentence is highlighted in yellow. Line 181.

  • 2- The plant identifier should be mentioned.

Response: Thank you very much for your suggestion, the plant was identified by the botanist Lahcen Zidane, professor at the Faculty of Science of Kenitra, University Ibn Tofail, Kenitra, Morocco. Line 193-194.

  • 3- A voucher specimen number should be mentioned.

Response: Thank you very much for your suggestion, the plant was identified by the botanist Lahcen Zidane, professor at the Faculty of Science of Kenitra, University Ibn Tofail, Kenitra, Morocco,. Line 193-194. And the specimen voucher number is MY002

  • 4- If allowed, the tested extracts should be subjected to further phytochemical analysis like HPLC-DAD-ESI-MS/MS analysis followed by molecular modelling studies like docking.  

Response: Thank you very much for your suggestion. We have taken your suggestion in consideration, The tested extracts were sent to the National Center for Scientific and Technical Research of Morocco for phytochemical analysis by HPLC/MS, and the results are not yet available.

Conclusion

  • 1- Conclusion should be supported by the promising results.  

Response: Thank you very much for your suggestion. We have taken your suggestion in consideration, the added paragraph is highlighted in yellow. Line 297-306.

Abbreviations:

  • List of abbreviations should be inserted by the end of the manuscript before references.

Response: We have taken your suggestion in consideration, this added part is highlighted in yellow. Line 27-338

References:

  • Scientific names of plant names and microbial strains should be typed in italic font and in proper font.

Response: We have taken your suggestion into consideration, the references have been corrected.

Reviewer 3 Report

Authors present In vitro Studies on the Antimicrobial and Antioxidant Activities of Total Polyphenols of Cynara humilis L. The paper has novelty aspect (as the authors stated it) but the manuscript must be much better structured and developed. Please see bellow my suggestions:

Keywords must reflect the main characteristic words of the paper (usually reflected also in the title) in the best way to increase the paper's relevance and chances to be found when searching it after key words. So, for the actual title, I suggest the following  keywords: Cynara humilis L.; antimicrobial activity; antioxidant activity; in vitro; total polyphenols; 2.2-diphenyl-1-picrylhdrazyl. (with semicolon between them, not with comma).   1.Introduction section is a single paragraph. Please separate it in 3-5 paragraphs, according to each idea developed.   Nowhere in references 1-4 is not about ancient use of medicinal plant. Please check and insert Bungau S.G., Popa V.-C. Between religion and science: some aspects: concerning illness and healing in antiquity, Transylv. Rev., 26(3), 2015, 3-19.   L71-73. Please make the aim of the study  a SEPARATE, LAST paragraph of this section, to be easier visible and develop it better.   2. Results. In any scientific paper, information must be given only once/in a single form - usually the most relevant / easy to understand:
  • Paragraph 76-82 is repetitive/in duplicate with Figure 1. Chose either the text, either the graphical form. Moreover, numerical values must be corrected to english style (with point, not with comma).
  • Table 1 is also repetitive/in duplicate with the text above it. For 2 numerical values is not necessary a table - please remove it, as it is not relevant at all.
  • Table 2. Same repetition with the text above it. Chose the text or the table. 
  • I suggest checking all the tables in this regard and proceed consequently.

3. Materials and methods. A new first section must be inserted, namely 3.1. Reagents and apparatus. Here, please check and provide the Model, Producer/manufacturer, City and Country for each apparatus used in the research, and the Producer and Country for each reagent/chemical used in the experimental part. Maybe the authors can make a table with all reagents/chemicals used and their CAS number, detailing in the same table the initial purity, concentration used or any other aspect they consider.

All the following subsections of section 3 must be renumbered.

4.Discussion section is totally missing. It is mandatory inserting and developing it.  Please compare your obtained results with literature data, maybe in a Table having as the last column Ref. (references). Additionally, at the final of Discussion, do not forget highlighting in a separate paragraph the strengths and the weakness (if there is one)  of the present study.

Renumber Conclusion section as 5.

References:

  • please check and provide for each of them all the requested information by the Plants Instructions for authors;
  • many of them are older than 10-15 years; I doubt there is no any more recent published papers related to some aspects you have presented in this manuscript.

Author Response

Response to Reviewer #3:

Authors present In vitro Studies on the Antimicrobial and Antioxidant Activities of Total Polyphenols of Cynara humilis L. The paper has novelty aspect (as the authors stated it) but the manuscript must be much better structured and developed. Please see bellow my suggestions:

  • Keywords must reflect the main characteristic words of the paper (usually reflected also in the title) in the best way to increase the paper's relevance and chances to be found when searching it after key words. So, for the actual title, I suggest the following  keywords: Cynara humilis L.; antimicrobial activity; antioxidant activity; in vitro; total polyphenols; 2.2-diphenyl-1-picrylhdrazyl. (with semicolon between them, not with comma). Lg

Response: We have taken your suggestion in consideration the added sentence is highlighted in yellow. Line 39-40.

  • Introduction section is a single paragraph. Please separate it in 3-5 paragraphs, according to each idea developed.   Nowhere in references 1-4 is not about ancient use of medicinal plant. Please check and insert Bungau S.G., Popa V.-C. Between religion and science: some aspects: concerning illness and healing in antiquity, Transylv. Rev., 26(3), 2015, 3-19.  

Response: We have taken your suggestion in consideration, the added paragraph is highlighted in yellow. Line 46-48

  • L71-73.  Please make the aim of the study  a SEPARATE, LAST paragraph of this section, to be easier visible and develop it better.

Response: We have taken your suggestion in consideration, the added paragraph is highlighted in yellow. Line 76-80

  1. Results. In any scientific paper, information must be given only once/in a single form - usually the most relevant / easy to understand:
  • is repetitive/in duplicate with Figure 1. Chose either the text, either the graphical form. Moreover, numerical values must be corrected to english style (with point, not with comma).

Response: We have taken your suggestion in consideration, the text is removed and the graphic form is preserved. Numerical values have been corrected in English style. Line 83-84

  • Table 1 is also repetitive/in duplicate with the text above it. For 2 numerical values is not necessary a table - please remove it, as it is not relevant at all.

Response: according to the other reviewers, the table and the text are preserved.

  • Table 2. Same repetition with the text above it. Chose the text or the table. 

Response: according to the other reviewers, the table and the text are preserved.  

  • I suggest checking all the tables in this regard and proceed consequently.

Response: according to the other reviewers, the table and the text are preserved.   

  • Materials and methods. A new first section must be inserted, namely 3.1. Reagents and apparatus. Here, please check and provide the Model, Producer/manufacturer, City and Countryfor each apparatus used in the research, and the Producer and Country for each reagent/chemical used in the experimental part. Maybe the authors can make a table with all reagents/chemicals used and their CAS number, detailing in the same table the initial purity, concentration used or any other aspect they consider.

Response: We have taken your suggestion in consideration, the added paragraph is highlighted in yellow. Line 193-197.

  • All the following subsections of section 3 must be renumbered.

Response: We have taken your suggestion in consideration,

  • Discussionsection is totally missing. It is mandatory inserting and developing it.  Please compare your obtained results with literature data, maybe in a Table having as the last column Ref. (references). Additionally, at the final of Discussion, do not forget highlighting in a separate paragraph the strengths and the weakness (if there is one)  of the present study.

Response: The two sections are combined in one (results and discussion). Line 81

  • Renumber Conclusion section as 5.

Response: We have taken your suggestion under consideration. Respecting the request of the reviewer 2, the conclusion number will be 4.

References:

  • please check and provide for each of them all the requested information by the Plants Instructions for authors;
  • many of them are older than 10-15 years; I doubt there is no any more recent published papers related to some aspects you have presented in this manuscript.

Response: we have taken your suggestion in consideration.

Round 2

Reviewer 3 Report

The authors responded to some of my suggestions but ignored the most important ones, about the content and the way of presentation.  The manuscript is poorly revised. I insist in the following aspects:

2. Results. In any scientific paper, information must be given only once/in a single form - usually the most relevant / easy to understand:

  • Table 1 is also repetitive/in duplicate with the text above it. For 2 numerical values is not necessary a table - please remove it, as it is not relevant at all.
  • Table 2. Same repetition with the text above it. Chose the text or the table. 
  • I suggest checking all the tables in this regard and proceed consequently. NO other Reviewer given his accord for maintaing text and Table for the same information, despite the authors' statement that "according to the other reviewers, the table and the text are preserved." Scientifically is NOT correct and for a scientific paper is unacceptable.
  1.  

L192. Is Folin-Ciocalteu not Folin-ciocalteau. There are 2 names  there. And it needs to be explained in the parenthesis or a reference.

The authors just mentioned that" The two sections are combined in one (Results and discussion). Line 81".  No Discussion section is developed in that part. The title of the 2nd section is "and Discussion" but Discussion is missing!!! Inserting few references in that section it means nothing compared to what real discussions should be. The entire manuscript is incomplete without real Discussion.

Regarding Abbreviations, that part must be removed. Please check  Instructions for authors https://www.mdpi.com/journal/plants/instructions and apply them as it is mentioned: "Acronyms/Abbreviations/Initialisms should be defined the first time they appear in each of three sections: the abstract; the main text; the first figure or table. When defined for the first time, the acronym/abbreviation/initialism should be added in parentheses after the written-out form". Instructions for Authors are given to be respected , they are not optionally.

I suggest introducing appendix in the text, as it is a single Table, but also both revising the numerical values in English style and explaining each abbreviation used in the table under it.